# Allelochemical Activity of Eugenol-Derived Coumarins on *Lactuca sativa* L.

**DOI:** 10.3390/plants9040533

**Published:** 2020-04-20

**Authors:** Kamilla Pacheco Govêa, Rafaella Sueko Tomita Pereira, Mateus Donizetti Oliveira de Assis, Pâmela Ingrid Alves, Guilherme Andrade Brancaglion, André Eidi Toyota, José Vaz Cardoso Machado, Diogo Teixeira Carvalho, Thiago Corrêa de Souza, Luiz Alberto Beijo, Luciene de Oliveira Ribeiro Trindade, Sandro Barbosa

**Affiliations:** 1Environmental Biotechnology & Genotoxicity Laboratory (BIOGEN), Institute of Nature Sciences (ICN), Federal University of Alfenas (UNIFAL-MG), Alfenas MG 37130-001, Brazil; kaamilla.pacheco@hotmail.com (K.P.G.); rafasueko1407@gmail.com (R.S.T.P.); mateus_muzambinho@yahoo.com.br (M.D.O.d.A.); pamelaingridalves@hotmail.com (P.I.A.); ludeoliveira_1@yahoo.com.br (L.d.O.R.T.); sandro.barbosa@unifal-mg.edu.br (S.B.); 2Pharmaceutical Chemistry Research Laboratory (LQFar), Faculty of Pharmaceutical Sciences (FCF), Federal University of Alfenas (UNIFAL-MG), Alfenas MG 37130-001, Brazil; guiabrancaglion@gmail.com (G.A.B.); andre.toyota@yahoo.com.br (A.E.T.); jose__cardoso@hotmail.com (J.V.C.M.); diogotcarv@gmail.com (D.T.C.); 3Statistics Department, Institute of Exact Sciences (ICEx), Federal University of Alfenas (UNIFAL-MG), Alfenas MG 37130-001, Brazil; prof.beijo@gmail.com

**Keywords:** lettuce, phytotoxicity, cytotoxicity, genotoxicity, antioxidant enzymes, lipid peroxidation, bioactivity

## Abstract

Coumarins are widely distributed substances in plant species that promote phytotoxic effects, allowing them to be exploited as herbicides less harmful to the environment, since many invasive species have demonstrated resistance to commercially available products. The derived coumarins used in this study had not been tested in plant models and their effect on plants was unknown. The objective of this study was to evaluate the phytotoxic action of these coumarins in bioassays with *Lactuca sativa* L., in order to select the most responsive substance whose toxicity was best elucidated by chromosomal complement and enzymatic antioxidant metabolism studies. From the phytotoxicity assays, coumarin 8-methoxy-2-oxo-6-(prop-2-en-1-yl)-2H-chromene-3-carboxylic acid (A1), reported here for the first time, was selected as the most responsive and caused a reduction in the following parameters: number of normal seedlings, fresh biomass, root length and shoot length. Subsequent studies demonstrated that this coumarin is cytogenotoxic due to damage caused to the cell cycle and the occurrence of chromosomal abnormalities. However, it did not interfere with antioxidant enzyme activity and did not cause lipid peroxidation. The changes caused by coumarin A1 described herein can contribute to better understanding the allelochemical actions of coumarins and the potential use of these substances in the production of natural herbicides.

## 1. Introduction

Recently, many invasive species have demonstrated resistance to commercially available herbicides. For this reason, there is a growing interest in research and development of processes based on alternative products to control these plants, with the objective of minimizing environmental impacts caused by their indiscriminate use [1,2,3].

Coumarins are a large group of secondary metabolites derived from phenolic compounds, are produced mainly by higher plants, and may also occur in some fungi and bacteria [4]. There are reports about the bioactivity of coumarins that indicate these substances are a potential alternative to commercial herbicides, which is based on inhibiting germination and initial growth and development, as well as blocking mitosis in innumerable biotests [5,6,7].

Besides the coumarins found in secondary metabolites of plants, which can be isolated using chemical processes, these substances can be synthesized in the laboratory and there are several synthetic representatives and their respective effects reported in the literature [8,9,10,11]. The coumarins used in this study are derived from eugenol, a phenolic compound of secondary metabolism of some plant species. Eugenol can also be chemically isolated or synthesized, and its chemical structure can be modified in the laboratory, serving as an important precursor for derivatives with different bioactivity profiles [12,13,14].

Studies of germination parameters and initial seedling growth can verify the biological effects of substances, including coumarins, by elucidating possible phytotoxic effects on the morphophysiology of plant species [5,15,16,17]. Correlating these phytotoxicity studies with changes in the cell cycle and behavior of the chromosomal complement results in a deeper understanding of the mechanisms of action of coumarins, making it possible to verify cytogenotoxicity as a biological effect of these substances [18,19]. In addition, studies of the activity of the antioxidant enzymes superoxide dismutase (SOD), catalase (CAT), and ascorbate peroxidase (APX) can infer the occurrence of possible oxidative stress on plants exposed to coumarins and their derivatives. These analyses can be correlated to lipid peroxidation, increasing what is known about stress mechanisms. 

There are few works in the literature that correlate the main effects of coumarins on initial plant growth with cellular events related to physiological, genetic, and antioxidant metabolism changes [20,21]. Therefore, it is necessary to determine the biological effect of coumarins on plant biotests to contribute to future studies about using these substances as viable ecological alternatives to commercial herbicides. In addition, this study is at the forefront because the six eugenol-derived coumarins used have not been tested in plant models.

The main advantages of using *Lactuca sativa* L. as a plant model lies in the sensitivity of the species even in low concentrations of tested compounds, as well as the low research cost. In addition, *Lactuca sativa* has other peculiarities that favor its use—fast germination in approximately 24 hours, linear growth over a wide range of pH variation, low sensitivity to osmotic potentials, establishment of plant with approximately 21 days, small number of chromosomes (2n = 2x = 18), and presence of large chromosomes [22,23]. The last two characteristics cited are advantageous for cytogenetic analysis [18].

The objective of the present study was to evaluate the phytotoxic action of the six coumarins synthesized from eugenol on bioassays with *Lactuca sativa* L., in order to choose the most responsive molecule whose toxicity was best elucidated by chromosomal complement and enzymatic antioxidant metabolism studies of the biotest.

## 2. Results

### 2.1. Synthesis and Characterization of the Eugenol-Derived Coumarins

The coumarins A1–A6 were obtained by synthesis from eugenol using the Knoevenagel method in good yields (52%–96%) and in sufficient quantities to conduct the planned assays. All of the coumarins were properly purified using silica column chromatography and their identities were confirmed using infrared and ^1^H and ^13^C nuclear magnetic resonance spectroscopy techniques. The details about the synthesis, purification, and analysis protocols can be checked in the Appendix A. The coumarins A1, A3, A5, and A6 are described here for the first time. The coumarins A2 and A4 have been reported elsewhere [24,25], but as part of studies for other purposes.

### 2.2. Phytotoxicity Assay

The analysis conducted on the seventh day demonstrated that all the treatments had similar frequency of germinated seeds, except those exposed to coumarin 8-methoxy-2-oxo-6-(prop-2-en-1-yl)-2H-chromene-3-carboxylic acid (A1) at 800 µg mL^−1^ concentration, which had a 25% lower frequency of germinated seeds compared to the control (Table 1).

The data for germination speed index (GSI) and total number of seedlings are not provided because the statistical analysis showed there was no significant interaction for both parameters (*p* = 0.084 and *p* = 0.078, respectively). In other words, the GSI and total number of seedlings averages did not differ among themselves for any of the treatments. 

In relation to the number of normal seedlings, except for the control, there was a statistical difference between the coumarins for all concentrations and the treatment with coumarin A1 had the lowest number of normal seedlings at concentrations of 400 and 800 µg mL^−1^ (Table 2). Root system abnormalities were observed, where the primary root was atrophied and/or darkened (Figure 1).

For fresh biomass data, all the treatments presented a significant difference between the coumarins, except for the 50 µg mL^−1^ concentration. Coumarin A1 had the lowest fresh biomass at higher concentrations (Table 3).

Root length (RL) had the lowest values for seedlings exposed to coumarin A1, mainly for the higher concentrations, and the lowest value observed (1.97 mm) was for the 800 µg mL^−1^ concentration. Shoot length (SL) was less influenced by the treatments; however, coumarin A1 (800 µg mL^−1^) promoted a greater effect (6.48 mm) (Figure 2).

The phytotoxicity assay demonstrated that coumarin 8-methoxy-2-oxo-6-(prop-2-en-1-yl)-2H-chromene-3-carboxylic acid (A1) was the most responsive, since it caused the greatest phytotoxic effect on *Lactuca sativa* by drastically reducing the frequency of germinated seeds, number of normal seedlings, fresh biomass, root length, and shoot length parameters. Therefore, the subsequent assays were conducted only with coumarin A1.

### 2.3. Cytogenotoxicity Assay

The mitotic index (MI) analyses of coumarin A1 (8-methoxy-2-oxo-6-(prop-2-en-1-yl)-2H-chromene-3-carboxylic acid) revealed that the 50 and 100 µg mL^−1^ concentrations did not differ statistically from the control. However, the 200 and 400 µg mL^−1^ concentrations resulted in an approximately 45% reduction compared to the control, although they were equal among themselves; the 800 µg mL^−1^ concentration differed from the others because it reduced the MI by 74.5% compared to the control. It is possible to apply a nonlinear regression correlating concentrations and MI, obtaining a concentration-dependent effect (Figure 3A) using a quadratic model (R^2^ = 0.9513).

For chromosomal abnormalities, anaphase bridges, telophase bridges, stickiness, lost chromosomes, and c-metaphases were identified (Figure 4). Micronuclei were not encountered in any treatment. Compared to the control, the 50 µg mL^−1^ concentration increased the frequency of stickiness in 153.33% and of lost chromosomes in 300%, whereas it did not differ statistically for any one of the other abnormalities. All concentrations of 100 µg mL^−1^ or above did not differ from the control (Figure 3B). 

### 2.4. Assay of Antioxidant Enzyme Activity and Lipid Peroxidation

There was no statistical difference for superoxide dismutase (SOD) activity, catalase (CAT) activity and lipid peroxidation (*p* = 0.1110, *p* = 0.1050 and *p* = 0.3395, respectively) among the concentrations of coumarin A1 (Appendix A, Appendix A).

Compared to the control, ascorbate peroxidase (APX) had reduced activity for all concentrations of coumarin A1, although the concentrations did not differ among themselves (Figure 5).

## 3. Discussion

In studies conducted with coumarins, Suksungworn et al. [26] verified that isoscopoletin and umbelliferone did not affect germination of *Mimosa pigra*; however, Saleh et al. [27] studied the effect of 1,2-benzopyrone on germination of *Vicia faba* L. and demonstrated that frequency of germinated seeds was only lower for the higher concentration tested. In the present study, the frequency of germinated seeds analysis demonstrated that the 800 µg mL^−1^ concentration of coumarin A1 was the only treatment that presented enough toxicity to reduce germination, corroborating what was found in the cited studies. Although both studies [26,27] reported that the highest concentrations of coumarins reduced the germination speed index (GSI), in the present work GSI was not affected by the coumarins tested. For coumarins, the following mechanisms that inhibit or delay germination are reported by Samajdar et al. [28]: early inhibition of water absorption by seeds, which can delay or prevent the recuperation of a stable membrane configuration; interference in membrane functions and/or inhibition of O_2_ consumption; generation of reactive oxygen species (ROS) by influencing the membrane system; blocking the activation of peroxidases; and interference with genes in the testa and/or blocking the synthesis of gibberellins. Thus, there may have been interference in the frequency of germinated seeds of *L. sativa* by the 800 µg mL^−1^ concentration of coumarin A1 due to one or a combination of these mechanisms. 

The total number of seedlings was not affected, which demonstrates that germinated seeds can develop a root and shoot. However, many studies conducted with coumarins report root length inhibition and/or abnormalities in the root system [16,17,29,30], which was also observed in this study. The number of normal seedlings was drastically reduced by the coumarins tested for the 800 µg mL^−1^ concentration, which resulted in atrophy, darkening and/or anomalies. However, for coumarin A1 this phenomenon was observed starting at 100 µg mL^−1^. Besides causing anomalies, this coumarin at 800 µg mL^−1^ concentration also significantly reduced the length of the roots. Root length is reported as the main parameter where the toxic effect of coumarins is verified [8,17,26,29,31]; these substances have caused drastic reductions in root length compared to the control because they mostly target the root [32]. Lupini et al. [16] proposed that the inhibitory effect promoted by coumarins on the root system of *Zea mays* can be mediated by auxin, and Lupini et al. [16] confirmed that root development of *Arabidopsis thaliana* was influenced by the interaction of coumarin and polar auxin transport, which could have occurred with *Lactuca sativa* seedlings in this study.

Shoot length was less influenced by exposure to the coumarin derivatives. This was also found by Andrade et al. and Gusman et al. [15,33], who showed that plants exposed to other allelochemicals tended to maintain the shoot, whereas the root system was damaged by the compounds. The root is more sensitive to these compounds’ action compared to the shoot, since it is exposed for a longer period by being the first organ where the compounds act.

In addition, several authors [8,21,27] report a reduction in fresh biomass for plant species exposed to coumarins, mainly at higher concentrations, corroborating the results found in the present study. Saleh et al. [27] demonstrated that plant growth of *Vicia faba* was affected by the coumarin 1,2-benzopyrone because this substance interferes with the endogenous phytohormones indoleacetic acid (IAA), abscisic acid (ABA), and gibberellic acid (GA3). In the present study, coumarin A1 was responsible for the greatest reduction in shoot length and fresh biomass for the 800 µg mL^−1^ concentration, and this effect may be correlated with a reduction in the biosynthesis of the phytohormones cited or their degradation, once these phytohormones control the growth and regulation of plants. It represents an interesting topic for a future study.

For coumarin A1, the mitotic index (MI) showed a concentration-dependent reduction, corroborating the root length data, since tissue elongation is mainly dependent on cell division in the meristematic zone [34]. It was observed that the 800 µg mL^−1^ concentration had the lowest root length (91.10% lower than the control), which is justified by the drastic reduction of the MI (84.52% lower than the control) verified through the cytogenotoxicity analyses. The results obtained are congruent with those found by Yan et al. [34] and Yuksel and Aksoy [35], who, respectively, studied the effect of the coumarins umbelliferone and dafnoretin on the MI of *Lactuca sativa* and of 1,2-benzopyrone on *Lens culinaris*. They also reported a reduction in the MI that represents a concentration-dependent effect. Some coumarins with antimitotic activity have been reported in the literature as having the ability to induce apoptosis or stop the cell cycle in the G0, G1, S, or G2-M phase [36,37,38]. 

The control showed chromosomal abnormalities, probably due to the high frequency of cell division in short periods of time, as observed in *Lactuca sativa* cv. Babá de verão and discussed by Santos et al. [37]. Chromosomal abnormalities are directly related to the rate of cell division, as it increases the chance of observing anomalies due to increased cell proliferation. The 50 and 100 µg mL^−1^ concentrations did not differ statistically from the control for the MI, indicating that coumarin A1 does not interfere with cytokinesis at these concentrations. However, the 50 µg mL^−1^ concentration provided an increase in the frequency of two chromosomal abnormalities compared to the control: stickiness (153.33%) and lost chromosome (300%). It is evidence of this coumarin’s effect on the karyokinesis of *L. sativa*. A lower frequency of chromosomal abnormalities was detected for 100 and 200 µg mL^−1^ and its absence was observed for higher concentrations (400 and 800 µg mL^−1^) of coumarin A1, which is clearly explained by the low MI of *L. sativa* exposed to these concentrations. According to Leme and Marin-Morales [18], chromosomal breaks and bridges indicate a clastogenic action of the toxicant on the biotest, whereas lost and late chromosomes, stickiness, or c-metaphases result in aneugenic effects. Thus, coumarin A1 caused aneugenic effects in meristematic cells of *L. sativa*, evident due to the high frequency of stickiness and lost chromosomes promoted by the 50 µg mL^−1^ concentration [18,19].

The capacity of coumarins to induce chromosomal abnormalities and apoptosis and to stop cell cycle has been widely reported previously [26,39,40,41,42,43]. It is associated with its anticancer activity [41,42,43] and is advantageous to be used as bioherbicides [26,39,40]. Graña et al. [39] characterized scopoletin as an auxin-like herbicide due to its phytotoxic action in *Arabidopsis thaliana* by causing disorder in numerous physiological and metabolic processes, among them the disarrangement of microtubules on which cell division depends. Suksungworn et al. [26] verified that isoscopoletin and umbelliferone presented phytotoxicity in *Mimosa pigra* roots by causing ultrastructural damages which are characteristics of programmed cell death (PCD), and other authors correlated the PCD effect of coumarins to cell cycle arrest [39,40,41]. Thakur et al. [36] correlated effects in cell cycle to the ability of coumarins to inhibit kinase proteins (cyclin-dependent kinases (CDKs)), which are directly related to controlling the cell cycle; on the other hand, other authors reported coumarins as a microtubule-targeting agent [39,42]. 

In this study, coumarin A1 (8-methoxy-2-oxo-6-(prop-2-en-1-yl)-2H-chromene-3-carboxylic acid) disturbs the cell cycle by affecting the meristematic zone of the roots, which is evidence of its cytotoxicity and can be observed in the mitotic index and the chromosomal abnormalities detected. Considering the ability of coumarins to inhibit kinases, disarrange microtubules, and disorder numerous metabolic processes, it represents an interesting topic to be elucidated in future studies with greater depth, since interference in the cell cycle and/or metabolism is something desirable in allelochemical substances with bioherbicidal potential.

Quantifying lipid peroxidation and the activity of the enzymes superoxide dismutase (SOD), catalase (CAT), and ascorbate peroxidase (APX) demonstrated that coumarin A1 (8-methoxy-2-oxo-6-(prop-2-en-1-yl)-2H-chromene-3-carboxylic acid) does not promote oxidative stress in *Lactuca sativa*. Our results contrast those of Araniti et al. [21], who reported an increase of 48% for lipid peroxidation compared to the control in *Arabidopsis thaliana* exposed to 1,2-benzopyrone. El-Shora and El-Gawad [44] studied the effect of the foliar extract of *Portulaca oleracea* L., which is rich in coumarins, on *Cucurbita pepo* L. biotest. They observed a concentration-dependent increase in lipid peroxidation and SOD, CAT, and APX enzymatic activity and correlated the increase of these parameters to strong oxidative stress caused by the extract. *Lactuca sativa* also exhibited an increase in lipid peroxidation and/or activity of one or more antioxidant enzymes when under oxidative stress [23,34,45,46,47].

Moreover, Yan et al. [34] concluded that the coumarins umbelliferone and daphnoretin displayed distinct mechanisms of action to induce phytotoxicity in *Lactuca sativa*. They correlated the inhibitory effect of umbelliferone on root length as mainly dependent on the overproduction of ROS, triggering oxidative damage. Although coumarin A1 caused a reduction in APX activity, it is not enough to confirm an oxidative damage. Thus, it demonstrates that the phytotoxicity promoted by coumarin A1 does not occur from oxidative stress. On the other hand, the inhibitory effect of daphnoretin on *L. sativa* root length was correlated to its effects on cell division [34], which can be verified in this study. Therefore, the phytotoxicity promoted by coumarin A1 is mainly associated with disturbances in cell cycle and DNA damage. 

Numerous herbicides’ mechanisms of action are cited in the literature, basically by inhibition of different enzymes, photosynthesis, pigment biosynthesis, lipid synthesis, auxin transport, amino acid biosynthesis, microtubule formation, cell division, cellulose synthesis, and others [48,49,50]. A potential bioherbicide must cause inhibition of one or a set of the factors mentioned. In this study, the highest concentrations of coumarin A1 reduced the frequency of germinated seeds, number of normal seedlings, fresh biomass, root length, shoot length, and mitotic index. In addition, the 50 µg mL^−1^ concentration increased the frequency of chromosomal abnormalities (stickiness and missing chromosome), which demonstrates the aneugenic effect of coumarin A1. These parameters show that coumarin A1 can be exploited as a potential bioherbicide, since it probably acts by inhibiting or disturbing cell division. However, it is necessary to carry out more in-depth studies in order to fully elucidate its mechanisms of action. This study provides initial information about coumarin A1, which is reported for the first time here, supporting future studies.

In an ideal scenario, herbicides should be highly selective to plants and non-toxic to other organisms, act quickly and effectively in low doses and/or concentrations, be quickly degraded in the environment, and present low production and consumption costs. However, few products available on the market satisfy all these criteria [50]. Accordingly, it is possible to highlight another advantage of using coumarin A1 as a bioherbicide, that is, this substance acts quickly and effectively in low concentrations, as demonstrated in the present study. It is emphasized again that more in-depth studies should verify whether coumarin A1 meets any one or other of the abovementioned criteria.

## 4. Materials and Methods 

### 4.1. Synthesis and Characterization of the Eugenol-Derived Coumarins A1–A6

The six coumarin derivatives used in this study were synthesized from eugenol using the traditional method known as Knoevenagel condensation (Figure 6). In this study, we wanted to verify the influence of different chemical groups attached to the C-3 position by a carbonyl function. Therefore, this synthetic method was chosen since it allows researchers to obtain coumarins with the required substitution pattern. First, eugenol was converted into a formylated derivative. Then, this aldehyde was subsequently submitted to cyclocondensation reactions with the respective dicarbonyl compounds in the presence of a basic catalyst under heat, to furnish the coumarins A1, A2, A3, and A4. The coumarin A5 was obtained by a reduction of the nitro group in coumarin A4 with tin chloride. This amino-coumarin A5 was also used as the starting material to furnish the coumarin A6 by acylation with maleic anhydride. Part of the eugenol structure is maintained in the coumarins (unsaturated side chain) and the other subunit is involved in the so-called coumarin nucleus. The nature of the carbonyl side chain differentiates the six coumarin derivatives. These coumarins were purified and characterized as shown in detail in the Appendix A.

### 4.2. Phytotoxicity Assay

Preliminary assays indicated the need to use a substance that helps disperse the coumarins in the agar. This was solved by using Tween 80 at a concentration of 8%, which was determined to be non-interfering with biotest metabolism in pre-tests.

Each coumarin (A1–A6) was suspended in a solution of agar (7 g L^−1^) containing 8% Tween 80 to obtain a stock solution at a concentration of 800 µg mL^−1^. The pH was measured and adjusted to 5.8 to ensure agar solidification [51].

For the phytotoxicity assay, 30 seeds of *Lactuca sativa* L. cv. Babá de verão were placed in Petri dishes (7 cm diameter) containing 10 mL of agar solution (7 g L^−1^) plus 8% Tween 80 with different concentrations of the coumarins—50, 100, 200, 400, and 800 µg mL^−1^—and agar plus 8% Tween 80 as negative control. Three repetitions were carried out per treatment.

The treatments were placed in a BOD incubator (Ethik Technology, 411FPD, Vargem Grande Paulista, SP, Brazil), set at 24 °C and a photoperiod of 12 h, for 7 days. The parameters evaluated were the following: frequency of germinated seeds, germination speed index (GSI), total number of seedlings, number of normal seedlings, fresh biomass, root length, and shoot length. Evaluations were made on the seventh day after the start of the experiment.

The GSI was determined using the formula proposed by Chiapusio et al. [52]:(1)$$\mathrm{GSI}=\frac{N_{1}}{1}+\frac{\left(N_{2}-N_{1}\right)}{2}+\frac{\left(N_{3}-N_{2}\right)}{3}+\cdots +\frac{\left(N_{n}-N_{n-1}\right)}{n}$$
where *N1*, *N2*, *N3*, *Nn*, and *Nn−1* correspond to the number of germinated seeds in the first, second, and third, *n*, *n−1* evaluations, respectively, and *n* is the evaluation number.

For total number of seedlings and number of normal seedlings analysis, the germinated plant material that had a developing root and shoot was considered as a seedling. Additionally, the seedlings similar to the control that did not visually exhibit a toxic effect were considered normal. To analyze the root length and shoot length parameters, the 10 largest seedlings from each Petri dish were chosen and measured with a digital caliper (DIGIMESS® 150 mm, São Paulo, SP, Brazil).

Based on the results, the most responsive molecule was chosen, which was the one that caused the greatest toxic effect. The subsequent experiments were conducted with the selected coumarin.

### 4.3. Cytogenotoxicity Test

For the cytogenotoxicity evaluations of coumarin A1 (8-methoxy-2-oxo-6-(prop-2-en-1-yl)-2H-chromene-3-carboxylic acid), *L. sativa* seeds were submitted to the same environmental conditions as those previously described. Root tips were collected 24 h after the start of the experiment, fixed in Carnoy’s solution (3 absolute ethanol: 1 glacial acetic acid) and stored at −18 °C.

The cytological preparations were made using the method described by Ribeiro et al. [53] and Santos et al. [37]. For each treatment, 6000 cells were evaluated to determine the mitotic index (MI) and verify the occurrence of chromosomal abnormalities.

### 4.4. Assay of Antioxidant Enzyme Activity and Lipid Peroxidation

For the assay of enzyme antioxidant activity and lipid peroxidation of coumarin A1 (8-methoxy-2-oxo-6-(prop-2-en-1-yl)-2H-chromene-3-carboxylic acid), 50 seeds of *L. sativa* were submitted for 10 days under the same conditions described for previous assays, in order to obtain the biomass needed for extraction.

To extract the antioxidant enzymes, whole seedlings (200 mg) were macerated in liquid N₂ with 10 mg of polyvinylpolypyrrolidone (PVPP) and homogenized in 1.5 mL of extraction buffer containing the following: 375 µL of 400 mM potassium phosphate, 15 µL of 10 mM EDTA, and 75 µL of 200 mM ascorbic acid. The homogenized material was centrifuged at 13,000 rpm for 30 min at 4 °C, and the supernatant was collected for the enzymatic analyses of superoxide dismutase (SOD), catalase (CAT), and ascorbate peroxidase (APX), which was quantified from spectrophotometer readings (Biochron, Libra S22, Holliston, MA, USA). The final volume of the reaction for reading enzyme activity was 2 mL (in a glass cuvette). All readings were conducted in duplicate.

Quantifying SOD activity was conducted according to the following method proposed by Giannopolitis and Ries [54]: 560 nm and one unit of SOD activity defined as the quantity of the enzyme that inhibits nitro-blue tetrazolium (NBT) photoreduction by 50%. To quantify CAT activity, the following method presented by Havir and McHale [55] was used: 240 nm, every 15 s for 3 min, and one unit of CAT activity defined as the quantity of enzyme that catalyzes decomposition of 1 µmol min^−1^ of H_2_O_2_. The APX activity was quantified according to the following method proposed by Nakano and Asada [56]: 290 nm, every 15 s for 3 min, and one unit of APX activity defined as the quantity of enzyme that oxidizes 1 µmol min^−1^ of ascorbate.

Lipid peroxidation was determined by quantifying the thiobarbituric acid reactive species at 540 nm, as described by Buege and Aust [57].

### 4.5. Statistical Analyses

The phytotoxicity experiment was a randomized block design (RBD), with a 6 × 6 factorial (six coumarins and six concentrations) and three repetitions. The remaining experiments were a randomized block design (RBD) with six concentrations and three repetitions. The data obtained were submitted to a Shapiro–Wilk normality test, followed by an analysis of variance (ANOVA). The averages were compared with the Scott–Knott test at 5% significance. The analyses were conducted with the program R (version 1.1.383, RStudio, Inc., Boston, MA, USA). 

## 5. Conclusions

This is the first report on the allelochemical action for these six eugenol-derived coumarins. The allelochemical action was mainly evident for the 800 µg mL^−1^ concentration, which interfered with the initial growth of *Lactuca sativa* seedlings. Coumarin A1 (8-methoxy-2-oxo-6-(prop-2-en-1-yl)-2H-chromene-3-carboxylic acid) was responsible for the most phytotoxic effect, promoting a drastic reduction in number of normal seedlings, fresh biomass, root length, and shoot length, as well as frequency of germinated seeds. 

Coumarin A1 influenced the cell cycle, reducing the mitotic index starting at 200 µg mL^−1^ concentration, indicating its cytotoxicity. The greatest frequency of chromosomal abnormalities was observed for the 50 µg mL^−1^ concentration, demonstrating the genotoxic effect of coumarin A1. It promoted aneugenic effects in *Lactuca sativa*, which is evident due to the high frequency of stickiness and lost chromosomes. These effects are not correlated with possible oxidative stress because coumarin A1, despite reducing APX activity, did not promote lipid peroxidation nor change SOD and CAT activity. Thus, the phytotoxicity promoted by coumarin A1 is mainly associated with disturbances in cell cycle and DNA damage, not from oxidative stress.

The physiological and cytogenetic changes caused by the coumarin A1, described here, help explain the allelochemical action of coumarins and the potential of these substances to be used in the production of natural herbicides. The parameters analyzed here indicate that coumarin A1 can be exploited as a potential bioherbicide, since it probably acts by inhibiting or disturbing cell division. This study provides initial information about coumarin A1, supporting future studies that are necessary to fully elucidate its mechanisms of action. 

## Figures and Tables

**Figure 1 plants-09-00533-f001:**
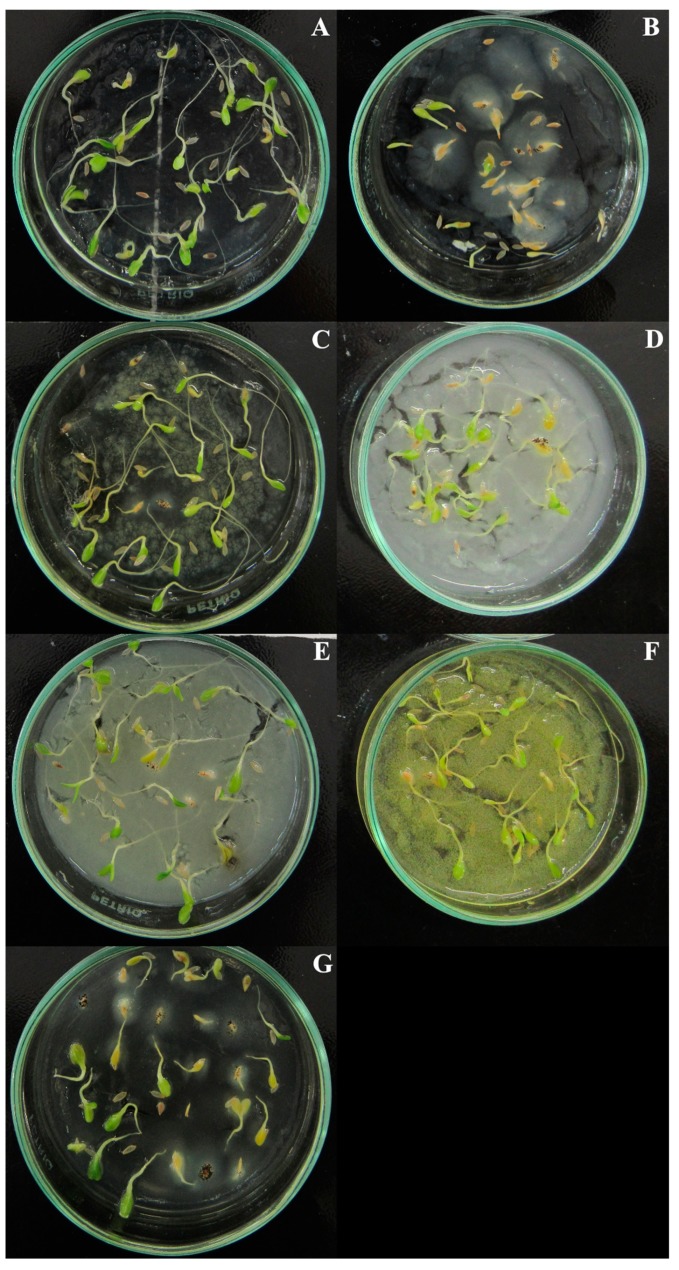
*Lactuca sativa* seedlings exposed to the six eugenol-derived coumarins at 800 µg mL^−1^ concentration, on the 7th day after the start of the experiment: (**A**) agar + 8% Tween 80 (control); (**B**) A1; (**C**) A2; (**D**) A3; (**E**) A4; (**F**) A5; (**G**) A6.

**Figure 2 plants-09-00533-f002:**
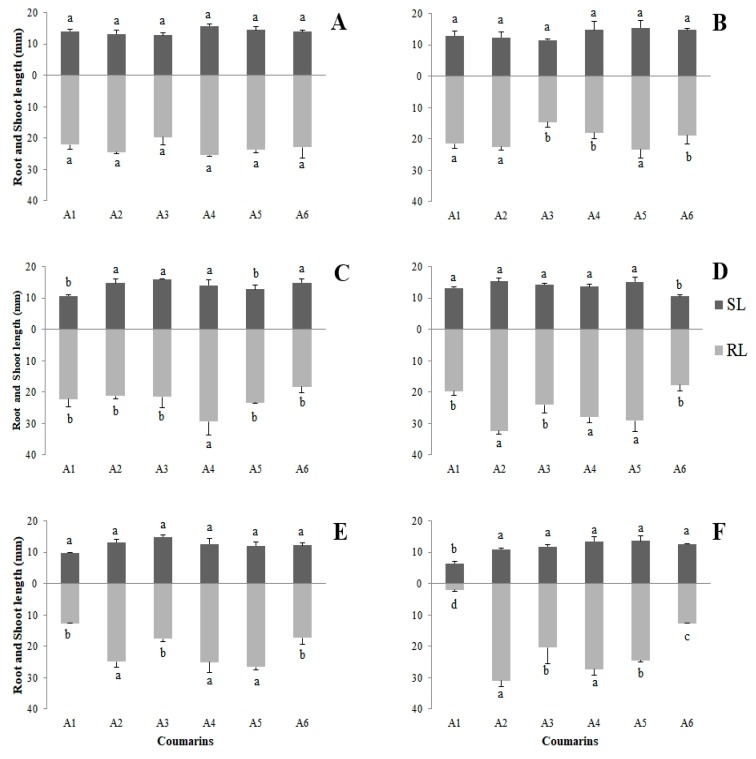
Averages obtained for root length (RL) and shoot length (SL), in millimeters, for *Lactuca sativa* seedlings exposed to the six eugenol-derived coumarins: (**A**) control; (**B**) 50 µg mL^−1^; (**C**) 100 µg mL^−1^; (**D**) 200 µg mL^−1^; (**E**) 400 µg mL^−1^; (**F**) 800 µg mL^−1^. Columns in the same color followed by the same letter do not differ statistically by the Scott–Knott test at 5% significance. Bar: standard error.

**Figure 3 plants-09-00533-f003:**
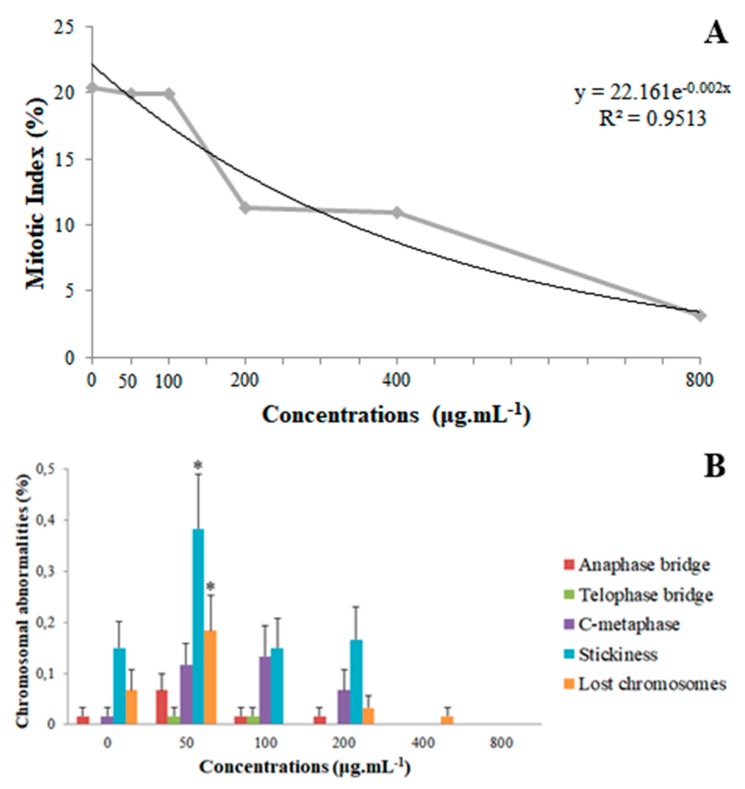
Cytogenotoxicity parameters. (**A**) Quadratic regression of averages of Mitotic index (MI) for the root tips of *Lactuca sativa* exposed to different concentrations of coumarin 8-methoxy-2-oxo-6-(prop-2-en-1-yl)-2H-chromene-3-carboxylic acid (A1). (**B**) Frequency of chromosomal abnormalities, in percentage, identified in the meristematic zone of *Lactuca sativa* roots exposed to the different concentrations of coumarin 8-methoxy-2-oxo-6-(prop-2-en-1-yl)-2*H*-chromene-3-carboxylic acid (A1). Columns with same collor followed by asterisks (*) differ statistically by the Scott–Knott test at 5% significance. Bar: standard error.

**Figure 4 plants-09-00533-f004:**
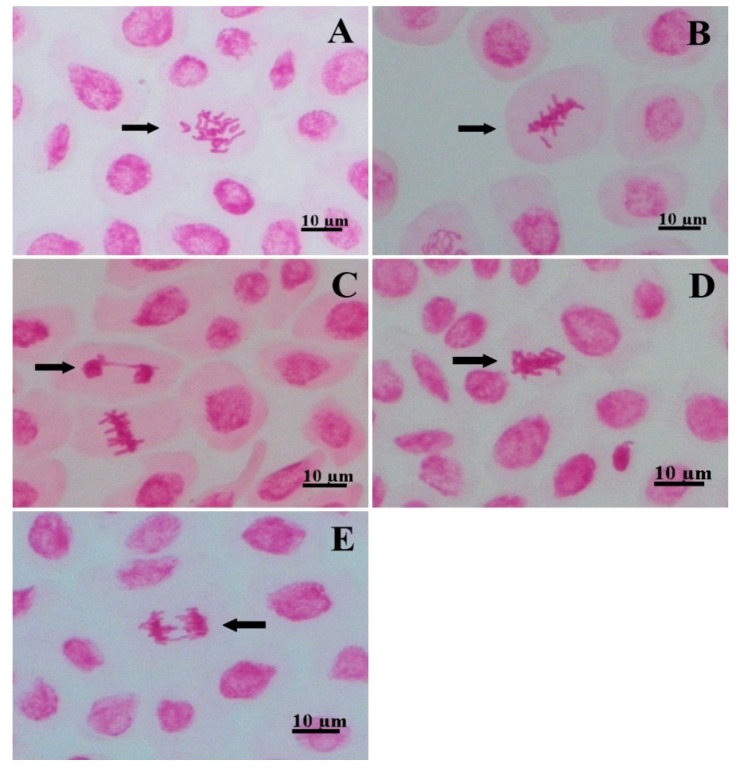
Chromosomal abnormalities identified in the root tips of *Lactuca sativa*: (**A**) C-metaphase; (**B**) lost chromosome; (**C**) telophase bridge; (**D**) stickiness; (**E**) anaphase bridge. Bar: 10 µm.

**Figure 5 plants-09-00533-f005:**
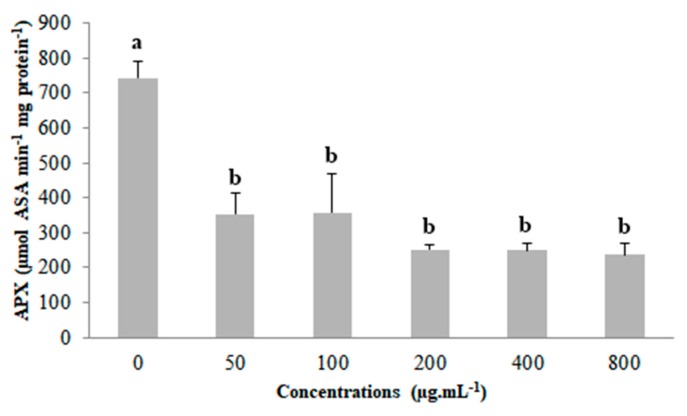
Ascorbate peroxidase (APX) activity in *Lactuca sativa* seedlings exposed to the different concentrations of coumarin A1. Columns followed by the same letter do not differ statistically by the Scott–Knott test at 5% significance. Bar: standard error.

**Figure 6 plants-09-00533-f006:**
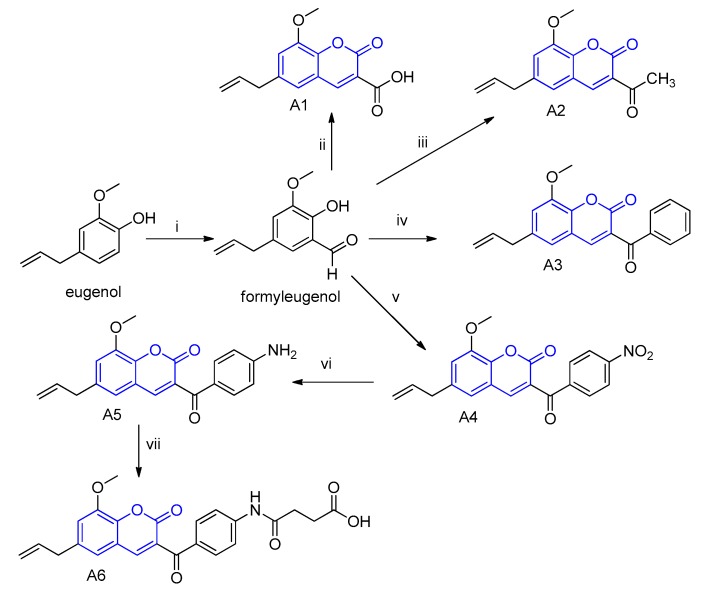
Synthetic route employed to obtain coumarins A1–A6 from eugenol: (i) hexamine, glacial acetic acid, 140 °C, followed by hydrolysis with aqueous HCl; (ii) malonic acid, piperidine, ethanol, 80 °C; (iii) ethyl acetoacetate, piperidine, ethanol, 80 °C; (iv) ethyl benzoylcetate, piperidine, ethanol, 80 °C; (v) ethyl 4-nitrobenzoylcetate, piperidine, ethanol, 80 °C; (vi) tin chloride, ethanol, reflux; (vii) succinic anhydride, pyridine, 80 °C. The coumarin chemical names are as follows: Coumarin A1*: 8-methoxy-2-oxo-6-(prop-2-en-1-yl)-2H-chromene-3-carboxylic acid*; Coumarin A2: *3-acetyl-8-methoxy-6-(prop-2-en-1-yl)-2H-chromen-2-one*; Coumarin A3: *3-benzoyl-8-methoxy-6-(prop-2-en-1-yl)-2H-chromen-2-one*; Coumarin A4: *8-methoxy-3-(4-nitrobenzoyl)-6-(prop-2-en-1-yl)-2H-chromen-2-one*; Coumarin A5: *3-(4-aminobenzoyl)-8-methoxy-6-(prop-2-en-1-yl)-2H-chromen-2-one;*
Coumarin A6: *4-((4-(8-methoxy-6-(prop-2-en-1-yl)-2H-chromen-3-carbonyl)phenyl)amino)-4-oxobutanoic acid*.

**Table 1 plants-09-00533-t001:** Frequency of germinated seeds of *Lactuca sativa* seedlings exposed to the six eugenol-derived coumarins.

Coumarin	Concentrations (µg mL^−1^)					
	0	50	100	200	400	800
**A1**	94.44% a	91.11% a	88.89% a	92.22% a	86.66% a	71.11% b
**A2**	90.00% a	91.11% a	88.89% a	93.33% a	97.78% a	91.11% a
**A3**	94.44% a	96.67% a	95.55% a	95.55% a	92.22% a	96.67% a
**A4**	94.44% a	84.44% a	95.55% a	93.33% a	88.89% a	93.33% a
**A5**	94.44% a	94.44% a	94.44% a	92.22% a	91.11% a	94.44% a
**A6**	96.67% a	93.33% a	93.33% a	90.00% a	91.11% a	87.77% a

Averages followed by the same letter, in a column, do not differ statistically by the Scott–Knott test at 5% significance.

**Table 2 plants-09-00533-t002:** Averages obtained for the number of normal seedlings of *Lactuca sativa* exposed to the six eugenol-derived coumarins.

Coumarin	Concentrations (µg mL^−1^)					
	0	50	100	200	400	800
**A1**	24.66 a	22.66 a	14.33 b	18.50 b	7.00 b	0.00 d
**A2**	24.50 a	24.00 a	25.00 a	25.50 a	24.00 a	15.33 b
**A3**	25.50 a	24.66 a	24.50 a	25.50 a	25.00 a	7.50 c
**A4**	24.33 a	19.00 b	24.00 a	25.33 a	24.50 a	18.66 b
**A5**	21.66 a	16.33 b	22.00 a	21.66 a	23.00 a	23.00 a
**A6**	22.66 a	22.33 a	25.50 a	15.00 b	22.00 a	1.33 d

Averages followed by the same letter, in a column, do not differ statistically by the Scott–Knott test at 5% significance.

**Table 3 plants-09-00533-t003:** Averages obtained for fresh biomass of *Lactuca sativa* seedlings exposed to the six eugenol-derived coumarins.

Coumarin	Concentrations (µg mL^−1^)					
	0	50	100	200	400	800
**A1**	0.42 a	0.27 a	0.18 c	0.18 c	0.15 d	0.14 c
**A2**	0.29 b	0.36 a	0.39 a	0.44 a	0.45 a	0.21 b
**A3**	0.25 b	0.25 a	0.46 a	0.33 b	0.37 b	0.38 a
**A4**	0.37 a	0.24 a	0.27 b	0.39 a	0.33 b	0.24 b
**A5**	0.41 a	0.29 a	0.40 a	0.32 b	0.25 c	0.32 a
**A6**	0.38 a	0.32 a	0.35 a	0.34 b	0.30 b	0.30 a

Averages followed by the same letter, in a column, do not differ statistically by the Scott–Knott test at 5% significance.

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
