# Peer review of "Allelochemical Activity of Eugenol-Derived Coumarins on Lactuca sativa L."

_plants, 2020, doi:10.3390/plants9040533_

Round 1

Reviewer 1 Report

The authors report on the synthesis and determination of allelochemical activity of six coumarin derivatives. The article is well-written but suffers from several issues which have to be addressed prior the final acceptance.

The authors should clarify the term “semi-synthetic” and explain better the context in which this term is used. Currently the title of manuscript is highly misleading,  suggesting that the coumarin derivatives analyzed therein are partly of natural origin,  whilst from the main text it is clear that all the derivatives tested were obtained synthetically with use of classical synthetic protocols.

The authors should revise the materials and methods section by adding the information on source and purity grade of all the chemicals and solvents used for both the synthesis and biological testing. In particular, this information should be implemented for eugenol, as it is unclear whether it was purchased from commercial vendor or isolated by the Authors from some plant material. If the eugenol was isolated from plant material please provide the information on plant species used and provide the detailed extraction procedure.

The logic behind the synthetic pathway designed should be clarified. In particular, the rationale/necessity  for Duff-formylation of eugenol step should be explained in detail. This is particularly important especially once considered that the eugenol-derived coumarins may be obtained via the von Pechmann mechanism. Please discuss the synthetic pathway designed and explain more clearly why the Knoevenagel mechanism was applied or why the series was not extended by 4-substituted coumarins deriving from the von Pechmann condensation.

Also, the comments on novelty of the derivatives obtained should be added. Currently the manuscript only contains an information that these derivatives were not tested for their allelochemical activity. In this context please specify whether these compounds are novel or were reported before. If the compounds are not novel please provide the suitable reference citations regarding their synthesis and biological activities determined to date.

The details provided in the Figure 6 caption are insufficient and the diagram for the synthesis given therein is inaccurate. Therefore the detailed synthetic protocols including the purification procedures should be implemented in the supplementary materials file. Particularly, the solvents used for the chromatographic purification should be clearly mentioned. Also the parameters of silica used for the columns should be given. The detailed information on yields of each compound (and especially for compounds A5 and A6) obtained should be provided. Based on the NMR spectra provided the compounds obtained were of excellent purity (and here I would like to congratulate the Authors on this achievement), hence the detailed information on the purification would be of particularly great value.

Based on the comments above I recommend the major revision of this manuscript.

Author Response

Dear Editor and reviewers of Plants,

We appreciate your contributions to our manuscript. All the topics listed were regarded relevant considerations and/or modifications that will greatly contribute to improving the quality of our article. The modifications proposed are listed and with detailed changes below, point by point. Alterations are highlighted (yellow) in the manuscript.

Reviewer 1:

  • “The authors report on the synthesis and determination of allelochemical activity of six coumarin derivatives. The article is well-written but suffers from several issues which have to be addressed prior the final acceptance.
  1. The authors should clarify the term “semi-synthetic” and explain better the context in which this term is used. Currently the title of manuscript is highly misleading, suggesting that the coumarin derivatives analyzed therein are partly of natural origin, whilst from the main text it is clear that all the derivatives tested were obtained synthetically with use of classical synthetic protocols.”

Answer: The term "semi-synthetic" refers to substances obtained by chemical synthesis based on compounds isolated from natural sources, such as microorganisms and plant species, as starting materials. The use of this term is also appropriate in where the natural product used is purchased from chemical suppliers, instead of having it isolated from plant species by the work authors. However, we agree with the statement you made when referring to the title of the manuscript, and we believe that its change to "Allelochemical activity of eugenol-derived coumarins on Lactuca sativa L." would make it reliable to what is reported although the text. So, we made this change in the title (line 2) in all the submitted files so that you can consider it. Therefore, there are some modifications in the Abstract (lines 22 and 23), Introduction (lines 50, 67-68 and 76-77), Results (lines 81-88) and Conclusion (line 414) sections, as well as in figures and tables titles (lines 97, 112, 117, 125 and 137).

  • “2. The authors should revise the materials and methods section by adding the information on source and purity grade of all the chemicals and solvents used for both the synthesis and biological testing. In particular, this information should be implemented for eugenol, as it is unclear whether it was purchased from commercial vendor or isolated by the Authors from some plant material. If the eugenol was isolated from plant material please provide the information on plant species used and provide the detailed extraction procedure.”

Answer: All information regarding the source and purity of reagents and solvents has now been added to the supplementary material (Supplementary file 1). The eugenol used was purchased from chemical suppliers and was not isolated from plants by us in the laboratory.

  • “3. The logic behind the synthetic pathway designed should be clarified. In particular, the rationale/necessity for Duff-formylation of eugenol step should be explained in detail. This is particularly important especially once considered that the eugenol-derived coumarins may be obtained via the von Pechmann mechanism. Please discuss the synthetic pathway designed and explain more clearly why the Knoevenagel mechanism was applied or why the series was not extended by 4-substituted coumarins deriving from the von Pechmann condensation.”

Answer: Our group chose to evaluate, in this first work, coumarin derivatives substituted in the C-3 position with carbonylated groups that showed differences in polarity, volume and ionization profiles. Thus, choosing the Knoevenagel method seemed to us the most convenient method. At that time, we were not interested in obtaining coumarins substituted in position C-4, so the Pechmann reaction was not employed, but it may be considered in a future work. This information regarding the choice of the structure pattern and, consequently, of the synthetic method, has now been properly inserted in the manuscript (Sections 2.1 and 4.1 – lines 81-88 and  316-343, respectively).

  • “4. Also, the comments on novelty of the derivatives obtained should be added. Currently the manuscript only contains information that these derivatives were not tested for their allelochemical activity. In this context please specify whether these compounds are novel or were reported before. If the compounds are not novel please provide the suitable reference citations regarding their synthesis and biological activities determined to date.”

Answer: Of these six substances described, there are those that have already been reported, whether in chemical synthesis or biological application and others that are new ones. Coumarins already known are: A2 and A4. The new coumarins are: A1, A3, A5 e A6. This information has now been properly inserted into the text with the suitable references (lines 26, 81-88).

  • “5. The details provided in the Figure 6 caption are insufficient and the diagram for the synthesis given therein is inaccurate. Therefore the detailed synthetic protocols including the purification procedures should be implemented in the supplementary materials file. Particularly, the solvents used for the chromatographic purification should be clearly mentioned. Also the parameters of silica used for the columns should be given. The detailed information on yields of each compound (and especially for compounds A5 and A6) obtained should be provided. Based on the NMR spectra provided the compounds obtained were of excellent purity (and here I would like to congratulate the Authors on this achievement), hence the detailed information on the purification would be of particularly great value.

Based on the comments above I recommend the major revision of this manuscript.”

Answer: We agree with your statements and we have changed the Figure 6 caption so that it can explain in details the information one often finds in synthetic route schemes (lines 333-343). We have now also inserted all the information regarding the synthesis, purification and characterization processes in the Supplementary file 1. Although not requested, we also insert the spectrometric data listings that confirm the identity of the products.

Reviewer 2 Report

The paper shows new findings about the phytotoxic action of six semi-synthetic coumarins, using one of the plant bioassays - Lactuca sativa L. test, in order to select the most responsive one as an herbicide. Chromosomal aberration tests and antioxidant metabolism studies were used. As a results Authors selected one of the coumarin, called A1 which showed clastogenic and aneugenic effects. These effects were not correlated with possible oxidative stress.  It is the most important result, unfortunately the Authors did not interpret it in order to possible using this coumarin as herbicide.

Other comments:

  1. Table 2 - title. The word ‘Averages’ seem to be inappropriate. I will recommend: ‘The frequency of germinated seedlings… ‘.
  2. Some unreasonable and difficult to perceive is to use different abbreviations to describe the effect of coumarins (NNS, G), since that they are not commonly recognized shortcuts.
  3. The frequency of chromosomal aberrations is very low (below 0.5%). In my opinion there is not enough to talk about these results as statistically significant. Especially that they are present in control material. The statistics is crucial.
  4. How Authors can explain the increased frequency of chromosomal aberrations after treatment with coumarin at the concentration 50 mml-1?
  5. What are RL data? (line 225)
  6. Cytogenotoxicity not citogenotoxicity (line 226)
  7. There is no damage of cell cycle, rather disturbances. DNA can be damage. (line 241)
  8. There is speculation about the role of CDK. It is not a subject of this study. (lines 233-243)
  9. ‘Intense mitosis’ – this no correct (line 244)
  10. Materials and methods 4.1.- the citation is necessary (line 275).
  11. General question - why Authors used Lactuca sativa test? there is no justification in the text.
  12. What is the mechanism of coumarin action to induce chromosomal aberrations?

General comments

The Authors should strongly indicate what is new and novel in their studies. Some practical recommendations and ideas should be pointed out in the Conclusions. Authors tried to solve the problem, however the interpretations of the results are sometimes not correct. Many results are too briefly described. The manuscript is poorly written and thus needs clarification of the language. Therefore, the manuscript is hardly understood. There are many sentences that require rephrasing for clarity.

Author Response

Dear Editor and reviewers of Plants,

We appreciate your contributions to our manuscript. All the topics listed were regarded relevant considerations and/or modifications that will greatly contribute to improving the quality of our article. The modifications proposed are listed and with detailed changes below, point by point. Alterations are highlighted (yellow) in the manuscript.

Reviewer 2:

  • “The paper shows new findings about the phytotoxic action of six semi-synthetic coumarins, using one of the plant bioassays - Lactuca sativa test, in order to select the most responsive one as an herbicide. Chromosomal aberration tests and antioxidant metabolism studies were used. As a results Authors selected one of the coumarin, called A1 which showed clastogenic and aneugenic effects. These effects were not correlated with possible oxidative stress. It is the most important result, unfortunately the Authors did not interpret it in order to possible using this coumarin as herbicide.”

Answer: We appreciate the suggestion, as this change will significantly improve the quality of the article. We agree with your note and chose to add or rewrite some excerpts from the Discussion (lines 287-314) and Conclusion (lines 420-427) sections. So we believe that we were able to clarify the phyto- and cytotoxic effects as capable of providing information that support the possibility of coumarin A1 being exploited as potential bioherbicides, although it is not correlated to an oxidative stress.

  • “1. Table 2 - title. The word ‘Averages’ seem to be inappropriate. I will recommend: ‘The frequency of germinated seedlings… ‘.”

Answer: Germination percentage (G%) is contained in Table 1; Table 2 refers to data of number of normal seedlings. We partially agree with the suggestion presented, since we comprehend that a reliable term would be “Frequency of germinated seeds”, not “seedlings”. The changes made can be observed in Results (lines 92, 94, 97 and 143-144), Discussion (lines 189-191 and 201), Materials and Methods (357-358), Conclusion (line 419) sections.

  • “2. Some unreasonable and difficult to perceive is to use different abbreviations to describe the effect of coumarins (NNS, G), since that they are not commonly recognized shortcuts.”

Answer: We appreciate this feedback and decided to remove the shortcuts all along the manuscript (lines 102, 104, 122, 143-145, 227, 232, 233, 243, 249, 364 and 367) , keeping only those notably known, such as MI (Mitotic Index) and GSI (Germination Speed ​​Index).

  • “3. The frequency of chromosomal aberrations is very low (below 0.5%). In my opinion there is not enough to talk about these results as statistically significant. Especially that they are present in control material. The statistics is crucial.”

Answer: We agreed with your recommendation and included the statistical analysis in the Results (lines 166-169) as well as presented a discussion about it (lines 243-251 and 254-256). We demonstrated that lower concentration of coumarin A1 (50 µg.mL-1) promoted an increase in the frequency of stickiness and lost chromosome compared to control (153.33% and 300%, respectively), indicating aneugenic effect of this coumarin. In addition, we have corrected Figure 3b, by adding the statistical information with asterisks (*) above the bars (lines 156, 161 and 162) as well as readjusted the Conclusion section (lines 421-424).

  • “4. How Authors can explain the increased frequency of chromosomal aberrations after treatment with coumarin at the concentration 50 mml-1?”

Answer: We are convicted that chromosomal abnormalities are directly related to the rate of cell division (reported here as mitotic index), as it increases the chance of anomalies due to increased cell proliferation. In our study, concentrations from 100 µg.mL-1  and above did not differ statistically among themselves nor from control in relation to chromosomal abnormalities. Since we have detected a cytotoxic concentration-dependent effect by analyzing mitotic index, 100 and 200 µg.mL-1 might presented similar numbers for  chromosomal abnormalities compared to control by reduction of mitotic index. A low frequency of chromosomal abnormalities was observed for higher concentrations (400 and 800 µg.mL-1) of coumarin A1, which is clearly explained by the low mitotic index of L. sativa seeds exposed to these concentrations. All this information can be verified in changes we have done in Results (lines 153-154 and 157) and Discussion sections (lines 243-251 and 254-256). In addition, we believe it is more reasonable to represent MI data by a non-linear regression, which can be seen in Figure 3a (line 156).

  • “5. What are RL data? (line 225)”

Answer: It represents “root length data” and was solved when we did the corrections proposed in item 2, related to the shortcuts.

  • “6. Cytogenotoxicity not citogenotoxicity (line 226)”

Answer: We have corrected this translation error and it can be verified on line 235.

  • “7. There is no damage of cell cycle, rather disturbances. DNA can be damage. (line 241)”

Answer: We agreed that it was provided an erroneous information. Although, this term was excluded of our discussion when we changed the paragraph (lines 243-244).

  • “8. There is speculation about the role of CDK. It is not a subject of this study. (lines 233-243)”

Answer: We agreed with the suggestion and removed excerpts that could be interpreted as speculation. We rewrote the paragraph so that it became clearer and more objective. It can be verified in lines 265-275.

  • “9. ‘Intense mitosis’ – this no correct (line 244)”

Answer: We have done a modification in sentence structure to correct this information (lines 241-242).

  • “10. Materials and methods 4.1.- the citation is necessary (line 275).”

Answer: We have edited the whole 4.1 Section as a recommendation of Reviewer 1. With done modifications, we believe that we were able to solve this question satisfactorily (lines 316-343).

  • “11. General question - why Authors used Lactuca sativa test? there is no justification in the text.”

Answer: The main advantages of using Lactuca sativa L. as a plant model lies in sensitivity of the species even in low concentrations of tested compounds, besides low research cost. In addition, Lactuca sativa has other peculiarities that favor its use: fast germination in approximately 24 hours; linear growth over a wide range of pH variation; low sensitivity to osmotic potentials; establishment of plant with approximately 21 days; small number of chromosomes (2n = 2x = 18) and presence of large chromosomes. The last two characteristics cited are advantageous for cytogenetic analysis. All these information were added to Introduction section (lines 69-75).

  • “12. What is the mechanism of coumarin action to induce chromosomal aberrations?”

Answer: Capacity of coumarins to induce chromosomal abnormalities are due to causing disorder in numerous physiological and metabolic processes, among them disarrangement of microtubules on which cell division depends. Some authors correlate coumarins’ phytotoxicity to ultrastructural damages – which are characteristics of programmed cell death – and cell cycle arrest or disorder. We thought it best to add this information to the Discussion section (lines 257-268), since we believe that it can clarify and improve quality of our study.

  • “The Authors should strongly indicate what is new and novel in their studies. Some practical recommendations and ideas should be pointed out in the Conclusions. Authors tried to solve the problem, however the interpretations of the results are sometimes not correct. Many results are too briefly described. The manuscript is poorly written and thus needs c”

Answer: We agreed that we could elucidate better the novel of our studies, and we did it in Abstract (line 26) and 2.1 section (lines 81-88). Some information was added or modified in Conclusion section in order to provide practical recommendations and ideas (lines 430-433) as pointed. In addition, some sentences were added or rewritten to clarify the study, as we agreed that we could make it more understanding (lines 93, 107-108, 121-123, 165-166, 200-202, 220-221, 229-230, 232-233, 234, 236-238, 241-243, 347-348, 351, 355, 359-360, 364-366, 375, 376, 417-418). Finally, we point out that the entire manuscript were thoroughly grammatically checked, so that the language was as clear and objective as possible.

We remain available for any clarifications.

With best regards,

Dr. Thiago Corrêa de Souza (*Corresponding author) & authors

Round 2

Reviewer 1 Report

Although the large number of issues was addressed properly in the revised version,  the manuscript still suffers from some shortcomings and there is a number of doubts which have to be addressed.

The supplementary file has not been updated whatsoever. Please provide an updated supplementary file, containing the detailed synthetic protocols as well as updated materials and methods information. Given the fact that without the updated supplementary file the issues present in first version of the manuscript have not been addressed.

These are as follows:

Issue number 2 -  The authors should revise the materials and methods section by adding the information on source and purity grade of all the chemicals and solvents used for both the synthesis and biological testing. In particular, this information should be implemented for eugenol, as it is unclear whether it was purchased from commercial vendor or isolated by the Authors from some plant material. If the eugenol was isolated from plant material please provide the information on plant species used and provide the detailed extraction procedure.”

Issue number 5 - ... “ the detailed synthetic protocols including the purification procedures should be implemented in the supplementary materials file. Particularly, the solvents used for the chromatographic purification should be clearly mentioned. Also the parameters of silica used for the columns should be given. The detailed information on yields of each compound (and especially for compounds A5 and A6) obtained should be provided.

Reviewer 2 Report

Manuscript was significantly improved. All responses to comments were included in the manuscript and authors exhaustively responsed to them. The manuscript has now a better form and thus has my recommendation to be published.